# Consumption of an Oil Palm Fruit Extract Promotes Large Bowel Health in Rats

**DOI:** 10.3390/nu12030644

**Published:** 2020-02-28

**Authors:** Michael A Conlon, Ravigadevi Sambanthamurthi, Yew Ai Tan, Kalyana Sundram, Syed Fairus, Mahinda Y Abeywardena

**Affiliations:** 1CSIRO Health & Biosecurity, Adelaide, SA 5000, Australia; michael.conlon@csiro.au; 2Malaysian Palm Oil Board, 6, Persiaran Institusi, Bandar Baru Bangi, Kajang Selangor 43000, Malaysia; raviga@mpob.gov.my (R.S.); syfairus@mpob.gov.my (S.F.); 3Malaysian Palm Oil Council, 2nd Floor, Wisma Sawit, Jalan Perbandaran, Kelana Jaya 47301, Selangor, Malaysia; kalyana@mpoc.org.my

**Keywords:** diet, oil palm, gastrointestinal tract, polyphenols, fibre, microbes, short chain fatty acids

## Abstract

Oil palm fruit is widely used for edible oils, but the health benefits of other components are relatively unknown. We examined if consuming a polyphenol-rich extract of the fruit, from a vegetation by-product of oil processing, which also contains fibre, has gastro-intestinal benefits in rats on a Western-type diet (WD). The oil palm preparation (OPP) was added to food (OPP-F) or drinking water (OPP-D) to provide 50 mg of gallic acid equivalents (GAE)/d and compared to effects of high amylose maize starch (HAMS; 30%) in the diet or green tea extract (GT; 50 mg GAE/d) in drinking water over 4 wk. OPP treatments induced some significant effects (*P* < 0.05) compared to WD. OPP-D increased caecal digesta mass, caecal digesta concentrations of total SCFA, acetate and propionate (OPP-F increased caecal butyrate concentration), the numbers of mucus-producing goblet cells per colonic crypt, and caecal digesta abundance of some bacteria which may provide benefit to the host (*Faecalibacterium prausnitzii*, *Akkermansia muciniphila* and *Ruminococcus gnavus*). HAMS induced similar effects but with greater potency and had a broader impact on microbe populations, whereas GT had minimal impacts. These results suggest dietary OPP may benefit the large bowel.

## 1. Introduction

The fruit of the African oil palm, *Elaeis guineensis*, is a significant global food source, being the world’s largest source of edible oils, and of high economic value to numerous countries [1]. The genome of this plant has recently been sequenced to enable a deeper understanding of its important traits [2]. The oil palm fruit mesocarp is a rich source of natural anti-oxidants, including carotene and vitamin E, and is likely to contain a range of beneficial compounds, including fibres, which have yet to be characterised or used, and an opportunity exists to identify bioactive components that may promote health, including health of the gastrointestinal (GI) tract.

There is growing evidence that the consumption of plant polyphenols can have many health benefits in humans [3] and oil palm fruit may be a rich source of these molecules. Dietary polyphenols are generally able to reach the large bowel, where they are metabolized by the resident bacteria, leading to the formation of bioactive metabolites which can contribute significantly to the effects of polyphenols [4]. Consequently, the ingested forms of polyphenols and their metabolites can come into contact with tissues in both the small and large intestine, thereby potentially impacting the health of those tissues. The health benefits of polyphenols are largely attributable to their immunomodulatory, anti-inflammatory and anti-oxidant properties [5,6], although green tea polyphenols may protect against cancers through both anti-oxidant and pro-oxidant mechanisms [7]. Epigallocatechin gallate (EGCG), a key green tea polyphenol, has been shown to protect against the formation of pre-neoplastic lesions in mice [8]. There are numerous other polyphenols from foods that have been shown to have gut health benefits, including those from pomegranates [9], olive oil [10] and grapes [11], to name just a few. Some of the benefits derived from consuming polyphenols may come from anti-microbial effects, particularly an ability to inhibit growth or activity of microbes responsible for infection [12].

Dietary fibres are important components of a healthy diet and the oil palm fruit is likely to be a rich source of dietary fibres, which have yet to be characterised or examined for their *in vivo* benefits. One of the most widely recognised benefits of dietary fibre is the strong association with the reduced risk of colorectal cancer (CRC) [13]. The beneficial effects on the large bowel can depend on the type of fibre consumed and include dilution of toxins, by increasing faecal bulk and increasing the numbers and activities of beneficial microbes, such as those which generate short chain fatty acids (SCFA). Resistant starch (RS), defined as starch which escapes digestion in the small intestine and reaches the large bowel, is a form of dietary fibre that is highly effective in stimulating the production of SCFA and as a consequence has multiple beneficial effects on gut tissues [14]. Experimental studies have demonstrated that dietary RS can protect against chemically-induced colorectal tumours and diet-induced colonic DNA damage and even protect against tissue damage and inflammation associated with colitis [15,16,17,18]. 

Recently, a novel preparation rich in polyphenols was extracted from the vegetation liquor generated during the milling and extraction of oil from the oil palm fruit [19]. This oil palm preparation (OPP) also contains fibre. In this study, we feed rats OPP to examine the potential of its constituents to benefit gut health. Specifically, we examine whether it has benefits in a model where rats are fed a diet moderately high in cooked red meat but low in fibre, factors which have previously been shown to result in poor colonic conditions, including higher levels of colonic DNA damage and reduced mucus barrier protection [17,20], which potentially underlies GI diseases such as CRC.

## 2. Materials and Methods

### 2.1. Animal and Diets

Fifty-six male Sprague-Dawley rats of ~200 g weight were obtained from the Animal Resource Centre, Murdoch University, Perth, Australia. They were housed in wire-bottomed cages in a room of controlled temperature (23 °C) and lighting (a 12 h light-dark cycle) and allowed free access to food and water. The rats were assigned randomly to 1 of 5 groups (*n* = 10–13 per group) and fed the experimental diets for 4 wk. The composition of experimental diets (Table 1) were modifications of the AIN-93 formulation [21] and contained 5 % wheat bran as a fibre source. Diets contained 48% cornstarch, except for the RS supplemented diet, which contained 18% cornstarch and 30% high amylose maize starch (HAMS; *Hi-maize*™, National Starch Food Innovation, Australia). To simulate a western-type diet (WD), all diets contained 25% cooked red meat (premium beef mince) and the fat used in the diets was sunflower oil (Crisco, Australia) which contained approximately 11% SFA, 20% MUFA, and 69% PUFA. The basal diet is referred to as the WD diet. OPP was added to the WD diet. OPP and green tea were also added to the drinking water and maintained at 50 mg gallic acid equivalents (GAE)/d (the volume was adjusted 3 times per week on the basis of the actual consumption to maintain the GAE dose). GAE was determined by assessing total phenol and other oxidation substrates and anti-oxidant content using the Folin–Ciocalteu method. OPP contained 3.5 g/100 g of total dietary fibre (TDF), as determined by a modification of AACC Method 32-05. That is, the homogenised sample (1 g) was treated sequentially with α-amylase at 100 °C, protease at 60 °C and amyloglucosidase at 60 °C to hydrolyse all of the starch and proteins in the sample. The fibre was precipitated by the addition of ethanol to make the final mixture 80% *v/v* ethanol. After standing overnight, the precipitated fibre was collected by filtration through a bed of Celite. The collected fibre was determined by weight after subtractions for ash, protein and enzyme blanks. Individual body weights were monitored throughout the study. At the completion of the dietary intervention period, rats were anesthetised with 4% halothane/oxygen to allow collection and weighing of gut tissues and digesta at the time animals were killed. Faecal and caecal digesta was homogenised and divided into aliquots for various analyses and frozen at −80 °C. Experimental procedures were approved by the animal ethics committee at CSIRO Health and Biosecurity and complied with the Australian code of practice for the care and use of animals for scientific purposes.

### 2.2. SCFA, Phenols, p-Cresol and Ammonia

Digesta was thawed and the liquid phase distilled, with heptanoic acid added as the internal standard (at 1.68 µmol/mL), followed by the separation of SCFA in the distillate, using gas liquid chromatography, as described previously [22]. The total SCFA was calculated as the sum of acetic, propionic, butyric, isobutyric, caproic, isovaleric and valeric acids. The methods used for the measurement of phenols, *p*-cresol and ammonia have also been described before [23,24]. Pools represent the total amount of a given component in the digesta collected from within the cecum or the colon. Levels are based on amounts in the wet weight of digesta.

### 2.3. Microbiology

DNA was extracted from thawed digesta by a process involving repeated bead beating in the presence of high concentrations of SDS, salt and EDTA, followed by purification using QIAamp columns [25]. Purified DNA was quantified using a Thermo Scientific Nanodrop 2000 Spectrophotometer (Thermo Fisher Scientific, Wilmington, DE, USA). The numbers of target bacteria were estimated using quantitative real-time PCR using a CFX Connect 96 real-time PCR detection system and CFX Manager v 2.1 (Bio-Rad, CA, USA). Details of the primers and PCR reaction conditions for each of the targets have been described previously [26]. An 8-series of 10-fold dilutions of a sample-derived standard containing the target amplicon were analysed in parallel with DNA samples for estimation of absolute abundance and PCR efficiency.

### 2.4. Histology

Colonic length was measured and 0.5 cm of tissue at the distal and proximal end was discarded. Samples for histological assessment (2 cm) were taken at 8 cm from the distal end. Colonic tissue was opened along the section length, cleaned of digesta, placed in histology cassettes, submerged in 10% neutral buffered formalin for 24 h and then stored in 70% ethanol until processed through graded levels of ethanol and chloroform before being embedded in wax. Embedded tissues were cut into 5 μm sections and were stained for mucins using Alcian blue (neutral mucins) and periodic acid-Schiff’s reagent (sulphomucins) in combination with previously described methods [27]. Histology reagents were supplied by Sigma. 

For wall thickness, 15–20 measurements were taken at different points along each section. Final wall thickness was reported as the average thickness (in µm) over the length of the wall. Crypts were measured from the base of the muscularis mucosa to the luminal surface. Crypt cell height was also recorded as the number of columnar epithelial cells on one side of the crypt. The total number of columnar epithelial cells (both sides of the crypt) were counted and recorded from an average of 8–10 crypts per section. Goblet cells in the crypt were counted and the total area of the goblet cells within each crypt was measured by drawing around each individual goblet cell and adding the areas together. The crypt cell area was calculated by drawing around individual crypts used for assessment. The total area of the goblet cells was expressed as a percentage of the crypt area. Goblet cells were recorded as containing either strongly sulphated mucins (pH < 2.5, staining purple/magenta) or neutral mucins (pH > 2.5, staining dark blue).

### 2.5. Statistics

All values are expressed as the mean ± standard error of mean (SEM). Comparisons were performed by one-way ANOVA followed by Tukey’s multiple-comparison post hoc test. Differences were considered to be significant at *P* < 0.05. Statistical analysis was performed using GraphPad Prism version 6.01 for Windows, GraphPad Software, La Jolla, California, USA.

## 3. Results

### 3.1. Body, Tissue and Digesta Weights

There were no significant differences in final body weight between the treatment groups (data not shown) or weights of organs (Table 2). 

Caecal digesta bulk (Table 2) was significantly increased relative to the WD group when OPP was consumed as a drink but not when added to the diet. The caecal bulk in rats consuming HAMS was significantly greater than in all other groups. Similarly, HAMS significantly increased the caecal tissue weights (Table 2) when compared to OPP and GT treatments.

The pH of the caecal digesta was lowered significantly relative to the WD group by HAMS but not by other treatments. In digesta from the colon, the pH was significantly lowered by HAMS relative to WD and both OPP treatments. Colonic digesta pH was significantly higher than WD for OPP drink (Table 2).

### 3.2. Impacts on Gastro-Intestinal Fermentation

Levels of individual and total SCFA in digesta (Table 3) were significantly influenced by treatment. In the cecum dietary, OPP and HAMS significantly increased total SCFA concentration relative to WD and GT. Relative to WD, acetate concentration was increased by HAMS, butyrate concentration was increased by dietary OPP, and propionate concentration was increased by dietary and drink OPP. When caecal SCFA pools were examined (the total amount present in the caecum) OPP drink and HAMS significantly increased total SCFA, acetate and propionate relative to WD and GT, with HAMS having significantly higher pools compared to all other treatments. In contrast, OPP diet (not drink) and HAMS significantly increased caecal butyrate pools relative to WD. SCFA levels in colonic digesta were also influenced by treatment. OPP drink significantly lowered colonic butyrate concentration relative to the WD treatment, but there were no other effects of the OPP treatments relative to the WD group. HAMS significantly increased colonic acetate concentration and lowered butyrate concentration relative to WD. When colonic digesta SCFA pools were calculated, only HAMS had a significant effect relative to WD, increasing pools of acetate, propionate and the total.

Protein fermentation products were also analysed in digesta and shown to be impacted by diet (Table 4). Ammonia concentration was significantly lowered by HAMS in both the caecum and colon when compared with the WD group. OPP drink significantly lowered colonic ammonia concentration relative to the green tea treatment. No treatments had effects relative to WD when caecal or colonic ammonia pools were calculated. Concentrations of phenols in the caecum were increased significantly by both OPP treatments and increased significantly in the colon by dietary OPP (with a trend of increased concentration by drink OPP). Concentrations of *p*-cresol were significantly lowered by HAMS relative to WD in the caecum and in the colon.

### 3.3. Bacterial Population Changes

Dietary treatment had a significant impact on the composition of bacterial populations in the caecal digesta, although total numbers of bacteria/g of digesta did not differ significantly between the treatments (Table 5). When rats consumed OPP as a drink, some significant effects on targeted microbial populations relative to the HR diet alone were observed. That is, OPP drink increased the numbers of *Faecalibacterium prausnitzii*, *Akkermansia muciniphila* and *Ruminococcus gnavus*. The positive control, HAMS, elicited the greatest number of changes in the microbiota, significantly increasing (relative to HR alone) the numbers of *Ruminococcus bromii,* the *Clostridium leptum* group, *Bifidobacterium*, *A. muciniphila* and *Ruminococcus torques*. The bacteria of the genus *Roseburia* decreased in number in response to HAMS. The numbers of *Lactobacillus* were increased by GT but not by OPP or by HAMS, and the numbers of *Bacteroides* were significantly higher for OPP drink when compared to GT.

### 3.4. Effects on Colonic Tissues

Histological analysis of colonic tissue (Table 6) revealed no changes in the thickness of the muscularis layer (data not shown) but some changes in the epithelial layer. The total number of cells and the number of mucus-producing goblet cells per colonic crypt was significantly higher for HAMS relative to all other treatments. OPP drink also significantly increased the number of goblet cells that produced neutral mucins per crypt compared to WD. When the goblet cell number was represented as a percentage of crypt cells, then only OPP drink significantly altered (increased) the percentage relative to WD. Appendix A shows example images of stained colonic tissues from each treatment group.

## 4. Discussion

This study examined whether a polyphenol-rich, fibre-containing extract from a vegetation liquor generated from the processing of the oil palm can have GI benefits in rats consuming a Western-type diet typically high in meat and fat. Results show that OPP consumption had a range of effects consistent with promotion of GI health. These include increased digesta mass, increased production of SCFA, shifts in digesta microbiota populations that are suggestive of benefit, and increases in mucus-producing goblet cells within colonic crypts. These effects were in many instances similar to those induced by dietary HAMS, a source of RS known to have multiple benefits to the large bowel, although OPP generally exhibited a lower potency. There were no obvious detrimental effects of OPP on gut health, although the consequences of the increase in colonic digesta pH are unknown. Consequently, there is potential for development of OPP or its constituents for use in humans to promote bowel health.

Consumption of dietary fibres has long been known to be associated with a reduced risk of colorectal diseases, especially cancer, and this reduced risk occurs through multiple mechanisms which can be dependent on the type of fibre. A primary means of benefit occurs as a result of the promotion of stool bulk, which acts to dilute toxins and assist their excretion [28,29,30]. Increased stool bulk occurs through a combination of the water holding capacity of the fibres and increased microbial mass resulting from the use of some fibres as substrates [28]. In this study, we showed that OPP increased digesta mass in rats, suggesting fibre-like effects, and that OPP may have stool bulking capability if it or a fibre-rich component were to be consumed by humans. 

Another mechanism through which dietary fibres may protect against colorectal disease is by the production of SCFA through microbial fermentation. Highly fermentable fibres, such as RS, can generate significant amounts of the main forms of SCFA, namely acetate, butyrate and propionate, and these in turn have multiple effects and benefits [14]. In addition to providing the main sources of energy for cells lining the colon [31], they also act to stimulate apoptosis of damaged cells [32], stimulate the immune system [33] and enhance gut barrier function via a mechanism that includes enhanced mucus production [34,35]. In the present study, OPP consumption resulted in greater SCFA production and increased the number of cells that produce mucus in the colonic epithelium, again suggestive of fibre-like activity and potential for protection against colorectal disease in humans. Dietary RS and other fibres have been shown to protect against colonic DNA damage induced by a Western-style diet in a rodent model [17], and there is evidence that they can protect against oncogenic processes in the colon of humans consuming a diet high in red meat, through a mechanism that may partly involve the SCFA [36]. However, in this study we did not see any evidence of a significant alteration in levels of colonic DNA damage with any of the treatments.

One of the most promising findings of the study relates to the effects of OPP on the composition of gut microbiota populations. There is a growing recognition of the important role that gut microbes play in GI health and also in mediating the health of tissues distant to the gut through mechanisms such as modulation of the immune system. Microbial products, such as butyrate, appear to have an important role in facilitating these effects [33]. Consequently, there is a significant global effort to understand the still relatively uncharacterized activities and roles of the many hundreds of microbial species within the gut [37] and to develop strategies to modulate the microbial populations through foods and dietary supplements. One such strategy is the use of prebiotics, defined as ‘selectively fermented ingredients that result in specific changes, in the composition and/or activity in the GI microbiota, thus conferring benefit(s) upon host health’ [38]. Many dietary fibres, including RS, may fit the description of prebiotics, not only because of their ability to increase the numbers of traditionally recognised bacterial markers of prebiosis (*Lactobacillus* spp. and *Bifidobacterium* spp.; also commonly used as probiotics) but also because of an ability to alter the numbers of other microbes with emerging roles in health [39]. Our study shows that OPP, like RS, can alter the numbers of some gut bacteria that were targeted for Q-PCR analysis because of their potential to impact gut health. OPP increased the numbers of *Faecalibacterium prausnitzii*, a key butyrate-producer with independent anti-inflammatory activity [40]. The numbers of some bacteria that appear to have important roles in gut mucus barrier turn-over (and potentially gut barrier integrity) were also increased by OPP. These bacteria, *Akkermansia muciniphila* and *Ruminococcus gnavus*, are low in the large bowel of individuals with inflammatory bowel disease [41,42] and in children with autism (many of whom have GI problems) [43]. Overall, HAMS had a slightly different and broader impact on microbial populations. Like OPP, HAMS increased the numbers of *A. muciniphila*. However, HAMS increased the numbers of a different *Ruminococcus* species associated with mucus, namely *Ruminococcus torques* [42]. Furthermore, unlike OPP, HAMS did not significantly increase the numbers of the butyrate-producer *F. prausnitzii* but did increase the numbers of the *Clostridium leptum* group, which contains many butyrate-producing bacteria. Some members of the genus *Roseburia* can also produce butyrate, and interestingly, the numbers of this genus were down in response to HAMS but not affected by OPP. HAMS increased the numbers of the starch degrader *Ruminococcus bromii*, as shown to occur previously in the stool of humans in response to dietary RS [44]. However, this species was not impacted by OPP. This suggests that the preparation does not contain appreciable amounts of RS. The numbers of other bacterial targets were not altered by OPP or HAMS, including sulphate-reducing bacteria (SRB; which produce toxic hydrogen sulphide), *Escherichia coli* and *Enterococcus faecium* (both potentially pathogenic), and the *Bacteroides* genus. The numbers of potentially beneficial *Lactobacillus* were increased by GT but not by OPP or by HAMS. Although this suggests that the polyphenols in GT but not OPP are able to stimulate *Lactobacillus* growth, it also confirms that dietary polyphenols can promote the growth of beneficial bacteria. 

While many of the effects we have observed in response to OPP suggest that fibre plays a central role, it is not possible to eliminate significant roles for polyphenols or other components of the OPP extract. We included GT as a treatment in this study, as it is a well-known source of polyphenols and has been demonstrated to have numerous beneficial biological impacts in vivo (5–7), including anti-microbial effects against some gut pathogens [45]. However, GT generally had little effect on the gut health markers examined when compared to the WD treatment and GT did not replicate the effects seen with OPP. Interestingly, only the GT treatment increased the numbers of *Lactobacillus*, suggesting some benefit to the gut. This indicates that the observed beneficial impacts of OPP on the large bowel are unlikely to be largely a result of polyphenols, although it is possible that GT and OPP polyphenols differ substantially in their characteristics and biological effects. Polyphenols derived from OPP may have benefits for tissues or activities that were not examined in this study and this should perhaps be explored in future work.

Levels of fibre in the OPP extract, and consequently in the respective treatments, are relatively low when compared to the amount of fibre given in the HAMS treatment, and this is likely to explain the lower potency of many of the beneficial effects seen with the OPP treatments. Purification of the fibre component from the OPP extract for future studies will allow a better assessment of the physical characteristics of the fibre(s) and allow more meaningful comparisons with HAMS and other beneficial fibres when examining physiological effects in vivo. The method of analysis used to determine the amount of fibre present in the OPP extract does not provide information on fibre type.

The effects observed suggest the route of OPP administration (food or drink) can influence physiological outcomes. Both routes of administration were designed to deliver similar doses of OPP on the basis of calculations of intakes and levels of inclusion for drink or food. The reasons for our finding of generally greater effects with the drink form of OPP administration can only be speculated upon. One possibility is that the incorporation of OPP into the food matrix hinders its availability to microbes and tissues within the large bowel to a greater degree than occurs for the liquid form. This should be taken into consideration in developing any dietary treatments that incorporate OPP or its constituents.

## 5. Conclusions

In conclusion, we provided evidence that the consumption of OPP can influence several markers of large bowel health that are indicative of benefits. Further studies to ascertain the constituents responsible for these benefits are warranted.

## Figures and Tables

**Table 1 nutrients-12-00644-t001:** Composition of experimental diets (g/kg diet or mg GAE/kg diet) and drinks ^1^.

Ingredient	WD	OPP-F ^2^	OPP-D ^3^	HAMS	GT
**Red meat**	250	250	250	250	250
**Cornstarch**	480	480	480	180	480
**Hi-maize ™**	0	0	0	300	0
**Sucrose**	100	100	100	100	100
**Sunflower oil**	70	70	70	70	70
**Wheat bran**	50	50	50	50	50
**l-Cystine**	3	3	3	3	3
**Choline bitartrate**	2.5	2.5	2.5	2.5	2.5
**Vitamins (ain-93)**	10	10	10	10	10
**Minerals**	35	35	35	35	35
**tert-butyl hydroquinol**	0.014	0.014	0.014	0.014	0.014
**OPP GAE/kg diet**	−	2000 mg	−	−	−
**OPP GAE in drink water**	−	−	50 mg GAE/d	−	−
**GT GAE in drink water**	−	−	−	−	50 mg GAE/d

^1^ Diets were based on AIN-93 (G) formulation. WD, Western diet control; OPP-F, OPP added to feed; OPP-D, OPP added to drink; HAMS, high amylose maize starch added to feed; GT, green tea added to drink. ^2^ OPP was added to the base WD feed formulation and was calculated on feed consumption at 25 g/d to provide 50 mg gallic acid equivalents (GAE)/d. ^3^ OPP or green tea was added to the drinking water of rats receiving the WD feed formulation and adjusted every 3 d on the basis of water consumption rates to provide 50 mg GAE/d.

**Table 2 nutrients-12-00644-t002:** Effects of dietary treatment on final body, organ, gut tissue and digesta weights, and digesta pH ^1^.

Weights	WD	OPP-F	OPP-D	HAMS	GT
**Body weight, *g***	306 ± 9	318 ± 11	294 ± 11	287 ± 19	303 ± 9

**Caecum**					
**Tissue weight, *g***	0.93 ± 0.18	0.72 ± 0.03 ^a^	0.79 ± 0.05	1.18 ± 0.08 ^ab^	0.61 ± 0.02 ^b^
**Digesta weight, *g***	1.86 ± 0.17 ^ab^	2.18 ± 0.21 ^c^	3.42 ± 0.37 ^ade^	4.82 ± 0.60 ^bcef^	1.48 ± 0.11 ^df^
**Caecum pH**	7.94 ± 0.08 ^a^	7.82 ± 0.09 ^b^	7.68 ± 0.06 ^c^	6.91 ± 0.16 ^abcd^	7.69 ± 0.07 ^d^

**Colon**					
**Digesta weight, *g***	0.96 ± 0.16 ^a^	1.43 ± 0.19 ^b^	1.40 ± 0.20 ^c^	3.37 ± 0.50 ^abcd^	1.68 ± 0.08 ^d^
**Colon pH**	7.49 ± 0.08 ^ab^	7.85 ± 0.15 ^cd^	8.04 ± 0.09 ^aef^	6.44 ± 0.07 ^bdfg^	7.14 ± 0.06 ^ceg^

**Organ weights, *g***					
**Liver**	11.33 ± 0.41	11.90 ± 0.52	11.27 ± 0.68	11.06 ± 0.65	10.55 ± 0.31
**Heart**	1.17 ± 0.04	1.19 ± 0.04	1.14 ± 0.04	1.06 ± 0.05	1.20 ± 0.07
**Kidney**	1.95 ± 0.07	2.10 ± 0.06	2.02 ± 0.08	1.86 ± 0.11	1.97 ± 0.05

^1^ Values are mean ± SEM for *n* = 10–13 animals per group. WD, Western diet control; OPP-F, OPP added to feed; OPP-D, OPP added to drink; HAMS, high amylose maize starch in diet; GT, green tea added to drink. Common superscript letters in the same row indicate significant differences, *P* < 0.05.

**Table 3 nutrients-12-00644-t003:** Effects of dietary treatment on individual and total SCFA in caecal and colonic digesta ^1^.

SCFA	WD	OPP-F	OPP-D	HAMS	GT
**Caecum** **Concentration ^2^**					
**Acetate**	41.4 ± 3.6 ^a^	55.5 ± 4.4	55.5 ± 2.7	68.1 ± 4.9 ^ab^	40.1 ± 2.9 ^b^
**Propionate**	6.0 ± 0.7 ^ab^	9.2 ± 0.9 ^ac^	9.5 ± 0.6 ^bd^	7.8 ± 0.8	5.1 ± 0.4 ^cd^
**Butyrate**	4.0 ± 0.3 ^a^	6.2 ± 0.8 ^abc^	2.8 ± 0.3 ^bd^	3.5 ± 0.4 ^c^	5.1 ± 0.6 ^d^
**Total**	53.4 ± 4.5 ^ab^	73.3 ± 0.6 ^ac^	69.3 ± 2.9	79.7 ± 5.6 ^bd^	51.9 ± 3.6 ^cd^

**Caecum pool ^3^**					
**Acetate**	73.2 ± 6.2 ^ab^	124.0 ± 17.7 ^c^	188.4 ± 19.5 ^ade^	328.9 ± 44.6 ^bcef^	60.4 ± 7.2 ^df^
**Propionate**	11.4 ± 1.7 ^ab^	20.9 ± 3.4 ^c^	33.6 ± 4.3 ^ad^	37.9 ± 5.8 ^bce^	7.7 ± 1.0 ^de^
**Butyrate**	7.1 ± 0.7 ^ab^	13.8 ± 2.1 ^a^	8.9 ± 0.9 ^c^	17.1 ± 2.6 ^bcd^	7.9 ± 1.5 ^d^
**Total**	94.9 ± 8.4 ^ab^	164.1 ± 23.4 ^c^	235.9 ± 24.1 ^ade^	384.7 ± 51.4 ^bcef^	78.4 ± 9.6 ^df^

**Colon** **concentration**					
**Acetate**	32.9 ± 3.6 ^a^	35.0 ± 2.4 ^b^	26.2 ± 3.0 ^c^	48.5 ± 4.3 ^abcd^	28.5 ± 1.7 ^d^
**Propionate**	4.8 ± 0.4	6.5 ± 0.5 ^ab^	6.3 ± 0.4 ^c^	3.7 ± 0.6 ^bc^	4.4 ± 0.2 ^a^
**Butyrate**	3.7 ± 0.5 ^ab^	3.0 ± 0.3	1.6 ± 0.2 ^ac^	2.0 ± 0.4 ^b^	3.7 ± 0.3 ^c^
**Total**	42.8 ± 4.5	46.1 ± 2.9	35.9 ± 3.2 ^a^	54.2 ± 4.7 ^ab^	38.1 ± 1.8 ^b^

**Colon pool**					
**Acetate**	33.1 ± 7.6 ^a^	50.3 ± 8.1 ^b^	35.6 ± 6.7 ^c^	170.9 ± 29.6 ^abcd^	47.9 ± 3.8^d^
**Propionate**	4.8 ± 1.0 ^a^	9.8 ± 1.7	8.8 ± 1.4	11.8 ± 1.9 ^a^	7.4 ± 0.5
**Butyrate**	3.3 ± 0.6	4.4 ± 0.8	2.3 ± 0.4 ^a^	6.6 ±1.5 ^a^	6.2 ± 0.6
**Total**	42.8 ± 9.4 ^a^	67.1 ± 10.7 ^b^	49.3 ± 8.4 ^c^	189.3 ± 31.5 ^abcd^	64.0 ± 4.4 ^d^

^1^ Values are mean ± SEM for *n* = 10–13 animals per group. WD, Western diet control; OPP-F, OPP added to feed; OPP-D, OPP added to drink; HAMS, high amylose maize starch in diet; GT, green tea added to drink. ^2^ Concentration is expressed as µmol/g digesta. ^3^ Pool is expressed as µmol. Common superscript letters in the same row indicate significant differences, *P* < 0.05.

**Table 4 nutrients-12-00644-t004:** Effects of dietary treatment on phenols, cresols and ammonia in caecal and colonic digesta ^1^.

Measure	WD	OPP-F	OPP-D	HAMS	GT
**Caecum ammonia**					
**Concentration ^2^**	7.97 ± 0.47 ^a^	8.38 ± 0.99 ^b^	5.76 ± 0.45	4.19 ± 0.6 ^abc^	7.76 ± 0.62 ^c^
**Pool ^3^**	15.26 ± 2.10	17.58 ± 1.78	18.89 ± 1.87	20.89 ± 3.77	11.96 ±1.66

**Caecum phenols**					
**Concentration**	0.68 ± 0.04 ^ab^	6.04 ± 1.02 ^acd^	5.55 ± 0.60 ^bef^	0.52 ± 0.02 ^df^	0.63 ± 0.02 ^ce^

**Caecum cresols**					
**Concentration**	5.71 ± 0.93 ^a^	8.12 ± 1.66 ^bc^	3.62 ± 0.77 ^b^	2.64 ± 0.49 ^acd^	6.90 ± 0.73 ^d^

**Colon ammonia**					
**Concentration**	10.59 ± 0.58 ^a^	9.06 ± 0.71 ^b^	6.59 ± 0.47 ^d^	5.10 ± 0.55 ^abc^	12.19 ± 0.31 ^cd^
**Pool ^3^**	12.70 ± 1.82	13.89 ± 1.6	9.34 ± 1.56 ^ab^	17.67 ± 3.00 ^b^	20.53 ±1.17 ^a^

**Colon phenols**					
**Concentration**	0.96 ± 0.07 ^a^	3.02 ± 0.34 ^abcd^	1.65 ± 0.17 ^be^	0.5 ± 0.01 ^de^	0.83 ± 0.06 ^c^

**Colon cresols**					
**Concentration**	14.9 ± 3.02 ^a^	9.98 ± 2.18 ^b^	13.74 ± 3.51 ^c^	1.54 ± 0.23 ^bd^	24.05 ± 3.4 ^acd^

^1^ Values are mean ± SEM for *n* = 10–13 animals per group. WD, Western diet control; OPP-F, OPP added to feed; OPP-D, OPP added to drink; HAMS, high amylose maize starch in diet; GT, green tea added to drink. ^2^ Concentration is expressed as µmol/g digesta. ^3^ Pool is expressed as µmol. Common superscript letters in the same row indicate significant differences, *P* < 0.05.

**Table 5 nutrients-12-00644-t005:** Effects of dietary treatment on numbers of caecal bacteria ^1^.

Bacteria	WD	OPP-F	OPP-D	HAMS	GT
***A. muciniphila***	8.92 ± 0.21 ^ab^	9.49 ± 0.28 ^c^	10.72 ± 0.20 ^bcd^	10.11 ± 0.14 ^a^	9.67 ± 0.20 ^d^
***Bacteroides***	10.54 ± 0.24	10.07 ± 0.34	11.23 ± 0.29 ^a^	10.70 ± 0.14	9.48 ± 0.45 ^a^
***Bifidobacterium***	9.84 ± 0.22 ^a^	9.09 ± 0.27 ^be^	9.53 ± 0.15 ^cf^	11.87 ± 0.16 ^abcd^	10.39 ± 0.22 ^def^
***C. coccoides***	7.83 ± 0.09	7.99 ± 0.09	8.18 ± 0.09	8.11 ± 0.08	8.05 ± 0.09
***C. leptum***	6.99 ± 0.13 ^a^	7.25 ± 0.11	7.25 ± 0.11	7.72 ± 0.12 ^ab^	7.19 ± 0.14 ^b^
***E. coli***	7.41 ± 0.23	7.02 ± 0.31	6.64 ± 0.34	6.51 ± 0.32	7.39 ± 0.17
***E. faecium***	6.20 ± 0.50	6.92 ± 0.77	6.54 ± 0.16	6.82 ± 1.06	6.38 ± 0.75
***F. prausnitzii***	10.29 ± 0.24 ^a^	10.15 ± 0.24 ^b^	11.65 ± 0.32 ^abc^	10.84 ± 0.24	10.51 ± 0.18 ^c^
***Lactobacillus***	7.13 ± 0.10 ^a^	7.33 ± 0.16 ^b^	7.44 ± 0.13 ^c^	7.62 ± 0.15	8.07 ± 0.12 ^abc^
***R. bromii***	9.18 ± 0.34 ^ab^	9.76 ± 0.27 ^c^	9.70 ± 0.21 ^d^	11.19 ± 0.28 ^acd^	10.35 ± 0.23 ^b^
***R. gnavus***	9.40 ± 0.13 ^a^	9.48 ± 0.11 ^c^	10.33 ± 0.09 ^abcd^	9.71 ± 0.10 ^b^	9.79 ± 0.18 ^d^
***R. torques***	8.76 ± 0.22 ^a^	9.13 ± 0.19 ^b^	8.75 ± 0.26 ^c^	10.48 ± 0.18 ^abcd^	9.18 ± 0.42 ^d^
***Roseburia***	7.04 ± 0.14 ^a^	6.71 ± 0.19	6.51 ± 0.08	6.22 ± 0.20 ^ab^	6.90 ± 0.14 ^b^
**SRB *dsr***	7.96 ± 0.15	8.25 ± 0.10	8.33 ± 0.08	8.12 ± 0.10	8.22 ± 0.15
**Total Bacteria**	11.63 ± 0.09	11.62 ± 0.12	11.62 ± 0.09	11.99 ± 0.09	11.76 ± 0.09

^1^ Values are expressed as log_10_ bacteria/g digesta (mean ±SEM) for *n* = 10–13 animals per group. WD, Western diet control; OPP-F, OPP added to feed; OPP-D, OPP added to drink; HAMS, high amylose maize starch in diet; GT, green tea added to drink. Common superscript letters in the same row indicate significant differences, *P* < 0.05.

**Table 6 nutrients-12-00644-t006:** Effects of dietary treatment on colonic tissue goblet cell numbers ^1^.

Histology Measure	WD	OPP-F	OPP-D	HAMS	GT
**Wall thickness (** **μ** **m)**	460 ± 29	400 ± 31	383 ± 34	366 ± 24	407 ± 60
**Total cells per crypt ^2^**	61 ± 2 ^a^	61 ± 2 ^b^	63 ± 1 ^c^	73 ± 2 ^abcd^	64 ± 2 ^d^

**Goblet cells per crypt**					
**Total**	6.3 ± 0.6 ^ab^	8.4 ± 0.5 ^ac^	9.5 ± 0.6 ^bd^	15.0 ± 0.6 ^bcde^	8.4 ± 1.0 ^e^
**Sulphomucin positive**	0.8 ± 0.2	1.1 ± 0.3	0.9 ± 0.4	0.7 ± 0.3	0.9 ± 0.3
**Neutral mucin positive**	5.5 ± 0.6 ^a^	7.3 ± 0.5 ^c^	8.6 ± 0.6 ^a^	14.3 ± 0.9 ^ace^	7.5 ± 1.0 ^e^
**Area (% of crypt)**	9.2 ± 0.9 ^a^	11.7 ± 0.8	13.8 ± 1.2 ^a^	12.5 ± 0.6	11.8 ± 1.7

^1^ Values are mean ± SEM for *n* = 10–13 animals per group. WD, Western diet control; OPP-F, OPP added to feed; OPP-D, OPP added to drink; HAMS, high amylose maize starch in diet; GT, green tea added to drink. ^2^ Total number of epithelial cells per crypt (8–10 crypts per section). Common superscript letters in the same row indicate significant differences, *P* < 0.05.

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
