# Peer review of "Consumption of an Oil Palm Fruit Extract Promotes Large Bowel Health in Rats"

_nutrients, 2020, doi:10.3390/nu12030644_

Round 1

Reviewer 1 Report

The authors of the manuscript "Consumption of an Oil Palm Fruit Extract Promotes Large Bowel Health in Rats" improved some of the comments from the first revision, namely in the introduction section and the details on the polyphenolic content of the palm oil extract. Even though, I still consider that the following aspects should be addressed before publication:

1. Considering the presence os polyphenols of the palm oil extract under study, the authors made an effort to compare the effect with a polyphenol rich beverage. The authors now present the polyphenol composition in ref 19 and clarified the quantification of polyphenols. Even though, the authors should also include the composition of green tea in order to provide a sound comparison between it and palm oil extract. Different polyphenols present distinct biological effects and thus, it is important to assess the similarities between palm oil and green tea composition.

2. Similarly, the authors compare the palm oil extract with a fibre-rich diet. In the discussion the authors consider that most of the health benefits of palm oil extract might be related to the fibre content. To support this claim, the authors should present the characterisation of the type of fibres contained in the palm oil extract under study. This is also crucial to clarify what is improving the GI parameters tested. Furthermore, the authors describe the health benefits of resistant starch, but hypothesise that its content might be low in this palm oil extract due to its inability to alter Ruminococcus bromii population. Thus, not only it is unclear what is improving GI health, but also seems incorrect to compare those effects with resistant starch benefits.

3. The inclusion of representative histological pictures is crucial to support the histological data presented by the authors. As part of the results presented in table 6, one representative image containing relevant measurement (the crypts, the goblet cell number and mucin staining) per treatment should be included. This would only amount to a total of 5 images.

Author Response

The authors of the manuscript "Consumption of an Oil Palm Fruit Extract Promotes Large Bowel Health in Rats" improved some of the comments from the first revision, namely in the introduction section and the details on the polyphenolic content of the palm oil extract. Even though, I still consider that the following aspects should be addressed before publication:

  1. Considering the presence os polyphenols of the palm oil extract under study, the authors made an effort to compare the effect with a polyphenol rich beverage. The authors now present the polyphenol composition in ref 19 and clarified the quantification of polyphenols. Even though, the authors should also include the composition of green tea in order to provide a sound comparison between it and palm oil extract. Different polyphenols present distinct biological effects and thus, it is important to assess the similarities between palm oil and green tea composition.

We thank the reviewer for their comment. Having the polyphenol composition of the green tea extract used in this study would be the ideal. However, the green tea extract used in the study, which was performed some years ago, was not assessed for polyphenol composition along with the OPP, and it is no longer available for analysis. Also, the key polyphenols present in green tea are well known and it would not be adding any truly novel information or insights. As indicated in the manuscript, the primary green tea polyphenols such as ECGC, which mediate the many benefits associated with its consumption, are well known. The Introduction states:

The health benefits of polyphenols are largely attributable to their immunomodulatory, anti-inflammatory and anti-oxidant properties [5, 6], although green tea polyphenols may protect against cancers through both anti-oxidant and pro-oxidant mechanisms [7]. Epigallocatechin gallate (EGCG), a key green tea polyphenol, has been shown to protect against the formation of pre-neoplastic lesions in mice [8].

Please also note, that the dose of OPP used in the study has been represented in terms of the equivalence of its potency to a key green tea polyphenol, namely gallic acid.

  1. Similarly, the authors compare the palm oil extract with a fibre-rich diet. In the discussion the authors consider that most of the health benefits of palm oil extract might be related to the fibre content. To support this claim, the authors should present the characterisation of the type of fibres contained in the palm oil extract under study. This is also crucial to clarify what is improving the GI parameters tested. Furthermore, the authors describe the health benefits of resistant starch, but hypothesise that its content might be low in this palm oil extract due to its inability to alter Ruminococcus bromii population. Thus, not only it is unclear what is improving GI health, but also seems incorrect to compare those effects with resistant starch benefits.

A future goal of our work is indeed to understand the role of OPP fibre in the observed effects described in the paper, and a part of that would be to obtain a detailed analysis of fibre types. It has only been through the study that we describe in the paper that it has been made clear that this is a path worth pursuing. Our initial goal was to examine the gut benefits of the OPP as a whole. As our findings suggest a range of effects/benefits which together are typically observed with fermentable fibres, our Discussion outlines this point (see excerpt below). We are not claiming that resistant starch is responsible for the observed benefits, indeed we indicate later in the Discussion that effects do not completely align with previously observed effects of resistant starch. We use comparisons to resistant starch as it was used a supplement in our study as it is a known fermentable fibre with multiple gut health benefits.

Results show that OPP consumption had a range of effects consistent with promotion of GI health. These include increased digesta mass, increased production of SCFA, shifts in digesta microbiota populations that are suggestive of benefit, and increases in mucus-producing goblet cells within colonic crypts. These effects were in many instances similar to those induced by dietary HAMS, a source of RS known to have multiple benefits to the large bowel.

We also point out in the Discussion that OPP components other than fibre, such as polyphenols, may also contribute to the gut effects seen.

In our final paragraph we highlight the need for future studies to ascertain which of the OPP constituents are responsible for the observed effects.

  1. The inclusion of representative histological pictures is crucial to support the histological data presented by the authors. As part of the results presented in table 6, one representative image containing relevant measurement (the crypts, the goblet cell number and mucin staining) per treatment should be included. This would only amount to a total of 5 images.

We have now provided some representative images of histology from each treatment group which illustrates the measurements taken. This is now provided as Supplementary Figure 1 (now referred to in section 3.4).

Reviewer 2 Report

I found the new version of the manuscript improved following the suggestions and comments given in the first revision. However, I strongly suggest changing every time you write “50 GAE” with “50mg GAE”. In the Material and Method section Line 88, I strongly suggest changing the sentence “GAE was determined using the Folin-Ciocalteau method” in “The total phenol content was determined using the Folin-Ciocalteau method (Reference)”.

Author Response

I found the new version of the manuscript improved following the suggestions and comments given in the first revision. However, I strongly suggest changing every time you write “50 GAE” with “50mg GAE”. In the Material and Method section Line 88, I strongly suggest changing the sentence “GAE was determined using the Folin-Ciocalteau method” in “The total phenol content was determined using the Folin-Ciocalteau method (Reference)”.

We have altered the wording in the Methods to indicate ‘GAE was determined by assessing total phenol and other oxidation substrates and anti-oxidants content using the Folin-Ciocalteu method’. Also, we have now used 50 mg GAE throughout the manuscript.

Reviewer 3 Report

A properly performed research aiming at identifying in a rat model possible gastrointestinal beneficial effects of an oil palm preparation (OPP) rich in fibers and polyphenols. Among the positive effects observed, of particular interest were the increase in mucus-producing goblet cells in the colonic epithelium and the relative increase as to control diet of the intestinal amount of some likely beneficial bacteria.

Major points:

in the opinion of this referee authors should significantly remodulate the presentation and in particular the discussion of the experimental findings achieved. In the present version of the manuscript, the most important experimental findings are not sufficiently highlighted; being OPP an extract containing a variety of compounds it is of course difficult to extrapolate mechanistic conclusions; still, authors should go deeper in this direction in a way to strengthen their report. At least by suitably and extensively discussing the possible mechanisms underlying the observed GI healthy effects of OPP;

Minor point: maybe a graphical abstract could further help in improving the overall presentation of this research outcomes.

Author Response

A properly performed research aiming at identifying in a rat model possible gastrointestinal beneficial effects of an oil palm preparation (OPP) rich in fibers and polyphenols. Among the positive effects observed, of particular interest were the increase in mucus-producing goblet cells in the colonic epithelium and the relative increase as to control diet of the intestinal amount of some likely beneficial bacteria.

Major points:

in the opinion of this referee authors should significantly remodulate the presentation and in particular the discussion of the experimental findings achieved. In the present version of the manuscript, the most important experimental findings are not sufficiently highlighted; being OPP an extract containing a variety of compounds it is of course difficult to extrapolate mechanistic conclusions; still, authors should go deeper in this direction in a way to strengthen their report. At least by suitably and extensively discussing the possible mechanisms underlying the observed GI healthy effects of OPP;

Minor point: maybe a graphical abstract could further help in improving the overall presentation of this research outcomes.

We thank the reviewer for their viewpoint. However, we do not understand which experimental findings are not sufficiently highlighted. We believe that we have provided enough discussion of the relevant outcomes and how they point to a need for deeper assessments and further health substantiation studies to progress the potential for gut health applications of the OPP. The study is not intended to deeply understand mechanisms, but rather provides evidence of gut effects and changes in gut health mediators which could be explored more deeply. Nevertheless, based on the available range of markers assessed, we have clearly outlined the potential for fermentable fibre-mediated effects of OPP via mediators such as SCFA and microbiota, and not excluded a role for other components such as polyphenols. It is not appropriate to speculate beyond this.

Hence, we do not agree with the need for a significant remodulation of the presentation and discussion of findings. We also do not believe that such a large-scale rewriting of the manuscript is feasible without withdrawing the paper from the current review process and making a new submission.

Round 2

Reviewer 1 Report

The authors of the manuscript "Consumption of an Oil Palm Fruit Extract Promotes Large Bowel Health in Rats" were able to improve the manuscript and clarified the reasons for not showing additional details on the polyphenolic content and fiber content of green tea and OPP respectively. Also, the inclusion of representative histological pictures strengthens the results presented in table 6. Overall, I believe that the current version of the manuscript should be considered for publication.

Reviewer 3 Report

I take note of the rebuttal letter by the Authors. I do not change my personal opinion. Ediotor should look for a third reviewer.

This manuscript is a resubmission of an earlier submission. The following is a list of the peer review reports and author responses from that submission.

Round 1

Reviewer 1 Report

General comments

The aim to assess plant preparations/extracts or any type of food matrices generally starts from the understanding their chemical composition. I think that this paper is incomplete from this point of view. Reading this manuscript, I honestly have a bit confusing ideas of where the effects come from. In which amounts the polyphenols are in the preparation? You say that you considered 50 GAE/kg diet for OPPF (not even described in the material and methods section how did you performed the assay for the total content of polyphenols as gallic acid equivalents). But you then compared with the green tea preparation, which can have different types and amounts of polyphenols (for example, do you have similar types of catechins in your OPP compared to the GT? What is the provenience of your green tea? is it a provided powder from a company?). In your patent, you describe the method of extraction, but can you please provide a link or some detailed data of HPLC quantifications of the polyphenol classes in the extract? I think this could help to better understand the observed effects.

Other many questions arise from the content of the fibers. How can you say that there are fibers and compare with the HAMS if you don’t show that the OPP contain them? How can I believe that good effects can come from a putative presence of fibers?

Specific comments

Line 16: “which also contains fibre” is it supported by a reference?

Line 35: Please provide a reference for the sentence “….to contain a range of beneficial compounds, including fibers, which have yet to be characterized or used”.

Line 51: “and grapes (particularly resveratrol)” I think this is not the case to extend the discussion to the resveratrol, unless you have resveratrol in your preparation. It seems to be too much speculating, just speak about flavonoids or phenolic acids, we all know that there are different classes of polyphenols with beneficial effects.

Line 67 “This oil palm preparation (OPP) also contains fiber”. Please provide a reference besides your patent.

Table 1. 50 GAE: can you please say if they are 50 mg or 50 g GAE or specify better what do you mean? This point is a bit unclear. Please describe the spectrophotometric assay (I guess) to complete the information about the polyphenol concentration, or please provide a reference for the method used for the determination. Have you followed the Folin-Ciocalteau method?

How did you estabilished 50 GAE/d for the dietary administration? Are there previous works in which 50 GAE have been demonstrated beneficial?

Line 216: Please provide a supplementary information in which you show pictures of the histological analysis of the examined tissues.

Line 240-247: it is too much speculation, please describe the type and amount of fiber you are talking of.

Line 281: I would avoid saying “…and in children with autism (many of whom have GI problems)”, because in your study you only analyzed the effect for potential intestinal problems without investigating the correlation with other diseases such as autism.

Reviewer 2 Report

The manuscript "Consumption of an Oil Palm Fruit Extract Promotes Large Bowel Health in Rats" investigated the benefits of the consumption of a palm oil preparation on the large bowel. Unfortunately, after carefully reviewing this manuscript, I found several flaws that make it unsuitable for publication in Nutrients, due to the following reasons:

1. The introduction can be improved by using more recent scientific publications. For example, green tea polyphenols not only act as direct antioxidants, they are currently known to modulate inflammatory and anti-oxidant signalling pathways like NF-kB and Nrf2, respectively.

2. Polyphenols are a large group of natural compounds and there is no description of the type of molecules the extracts contain. The type of molecules might depend on factors like the palm fruits growing conditions and processing techniques used for the extractions. Therefore, this paper would largely benefit from a characterization of the polyphenol content. The same applies to the fibre content.

3. The experimental design was not prepared correctly since all experiments were performed without a healthy control group. Therefore, it is impossible to conclude if the changes observed with the treatments represent a benefit towards an healthier state.

4. When presenting the effects on the colonic tissue, it would be relevant to present representative pictures of the colons morphology (macroscopic and histological pictures).

5. In the discussion section, lines 238-239, the authors consider that the OPP extract promotes bowel health without data or an hypothesis to back this assumption.

6. In line 260-261, the authors mention results that are not presented in the Methods nor in the Results section.